# Endogenous Ethanol Metabolism and Development of MASLD-MASH

**DOI:** 10.3390/ijms26178609

**Published:** 2025-09-04

**Authors:** Núria Farràs Solé, Sander Wydh, Amir Hossein Alizadeh Bahmani, Thi Phuong Nam Bui, Max Nieuwdorp

**Affiliations:** 1Department of Experimental Vascular Medicine, Amsterdam UMC, 1105 AZ Amsterdam, The Netherlands; n.farrassole@amsterdamumc.nl (N.F.S.); sanderwijdh@gmail.com (S.W.); a.h.alizadehbahmani@amsterdamumc.nl (A.H.A.B.);; 2Department of Internal and Vascular Medicine, Amsterdam UMC, 1105 AZ Amsterdam, The Netherlands; 3Diabeter Center Amsterdam, 1066 EC Amsterdam, The Netherlands

**Keywords:** MASLD, endogenous ethanol, fermentation, gut microbiome, alcohol dehydrogenase

## Abstract

Metabolic dysfunction-associated steatotic liver disease (MASLD) is an increasingly prevalent liver disorder driven by metabolic dysregulation and inflammation. Recent studies highlight the importance of the gut microbiome as a key contributor to this pathology through its ability to ferment dietary sugars into ethanol, a metabolite previously overlooked in MASLD. In this review, we discuss the role of the gut microbiome in MASLD, covering functional and compositional shifts observed in the disease; we dive into the different microbial pathways of ethanol synthesis, hepatic mechanisms of ethanol clearance, and pathological consequences. We also discuss the role of a healthy microbiome in the clearance of ethanol in the gut and how microbiome-based strategies could be beneficial in targeting endogenous production of ethanol, going from the traditional probiotic–prebiotic combination to discussing new approaches.

## 1. Introduction

The balanced composition and functionality of the gut microbiome (GM) is paramount in maintaining homeostasis in the host [1]. However, there has been a lack of consensus in the field regarding the definition of a healthy microbiome, which has led to commonly defining it as a compositionally diverse, functionally redundant and metabolically adaptable community [1,2]. The diversity of the GM presents inter-individual differences and is heavily influenced by socioeconomic and behavioral factors, such as diet, lifestyle and antibiotic use [3,4,5]. It has been described that a modern Western lifestyle defined as a high intake of processed foods added to a low-fiber and limited physical exercise, can drive towards an increase in metabolic dysfunction-associated steatotic liver disease (MASLD), type 2 diabetes (T2D), obesity and cardiovascular diseases [6]. Furthermore, a western diet has been associated with a decreased microbial diversity and an increase in pathogenic species, both of which have been linked to altered gut microbial metabolism and fat accumulation [7,8]. Contrarily, the ingestion of indigestible polysaccharides and fibers allows for their fermentation by gut microbes to produce mostly short-chain fatty acids (SCFAs), such as butyrate, propionate and acetate, and other fermentation products [9]. This type of metabolism from the gut microbiome has been linked to a decreased risk in metabolic diseases [10] due to the essential roles of SCFAs in the human body, including the maintenance of gut homeostasis, regulation of the epithelial gut barrier and immune response [9,11,12,13]. Overall, a balanced and metabolically diverse GM is crucial for the maintenance of a healthy gut environment and overall host homeostasis.

Dietary and lifestyle factors are not the sole determinants of the gut microbial compositions, and microenvironmental conditions within the intestinal tract have a crucial role in shaping microbial niches and the microbes that will thrive in them. Among local physiological variables, transit time, oxygen and pH are crucial factors influencing microbial growth [14,15,16]. Transit time has been described as a main driver for microbial profiles, influencing the microbial community at the taxonomical and functional levels [14]. Additionally, the oxygen gradient is a crucial ecological filter that allows for strict anaerobes, microaerophilic or facultative anaerobes to reside in a specific region of the gastrointestinal (GI) tract. The presence of oxygen falls steeply from the microaerophilic duodenal lumen, to near-anoxia in the cecum and colon, and within any given site drops from the lumen into the mucus layer that covers the epithelium [15]. This longitudinal and radial gradient allows facultative anaerobes to colonize the upper part of the GI tract, while obligate anaerobes reside in the anoxic regions of the lumen and large intestine [17]. In addition, intestinal pH is also an important physiological factor influencing the metabolic properties of gut microbes. Generally, this ranges from a more acidic environment in the upper regions of the GI tract and neutralizes towards the rectum, with a short acidic peak in the initial region of the colon [18]. However, individual differences and dietary metabolites and their degradation products influence the pH thought the gastrointestinal tract [16]. These physiological characteristics of the GI tract not only shape the microbial composition, but also the metabolic pathways which microorganisms are able to perform in a given region.

The study of how gut microbes are altered in MASLD is based on the impact of both internal and external variables on the gut microbial composition, diversity and function [19,20,21,22]. In the recent years, more and more studies have sprung and pointed towards the relevance of certain microbial shifts in MASLD while the mechanistic insights of these associations were not depicted [22,23,24,25]. However, recent advancements have emerged in the ability of the GM to produce ethanol via fermentative metabolism and its contribution to MASLD [20,26]. This review explores current knowledge on the metabolism of ethanol in the gut microbiome, discussing the processes involved in the synthesis of ethanol and potential clinical applications.

## 2. The Role of the Gut Microbiome in MASLD-MASH

MASLD and its inflammatory progression, metabolic dysfunction-associated steatohepatitis (MASH), are complex liver conditions characterized by steatosis and metabolic alterations. The progression of MASLD to MASH is characterized by hepatocyte ballooning, fibrosis and inflammation in the liver [27,28]. Additionally, the development of more severe conditions derived from the inflammation and altered metabolism can lead to advanced fibrosis, development of cirrhosis and hepatocellular carcinoma [28]. Evidence has shown that lipid metabolism in the liver is the main hallmark for MASLD, with de novo lipogenesis upregulation and impaired β-oxidation that result in an increased lipid accumulation in the cell and culminate in hepatocyte injury [29]. The effects of such alterations are linked to mitochondrial dysfunction, endoplasmic reticulum stress, and increased oxidative stress in the cell that derive from the metabolization of ethanol [29,30]. The interconnectedness between the gut and the liver via the portal vein explains the impact of diet-derived metabolites, such as ethanol, on liver function. In order to depict the implications and significance of the small and large intestinal microbiome in the synthesis of ethanol in MASLD patients, Meijnikman et al. demonstrated via postprandial sampling of the portal vein that higher ethanol concentrations were present in MASLD-MASH patients compared to healthy individuals after a high sugary drink intake [20]. Hence, although no ethanol is ingested in MASLD patients, a sugar-rich diet can be the cause of endogenous ethanol production and its effects in the liver.

### 2.1. Implications of the Metabolization of Ethanol in the Liver

The oxidation of ethanol in the liver is driven by an alcohol dehydrogenase (ADH) and under high-ethanol conditions by the cytochrome P450 2E1 (CYP2E1) in both the liver and intestinal epithelium, which leads to the formation of reactive oxygen species (ROS) and subsequent cellular damage at both sites [31,32,33]. The conversion of ethanol into acetaldehyde mediated by the peroxisome is considered a minor pathway in the metabolism of ethanol in the liver, although it also leads to ROS formation (Figure 1) [31]. Acetaldehyde is subsequently oxidized by an acetaldehyde dehydrogenase (ALDH) into acetate, which can be used by peripheral tissues as substrate in energy metabolism [31]. However, the presence of highly reactive acetaldehyde within the hepatocyte allows for it to generate cellular damage and trigger inflammation (Figure 1) [31,34]. Contrary to expectations, a recent study showed a reduced ADH activity in MASLD patients and diet-induced MASLD phenotype C57BL/6J mice [35]. The same authors linked this decreased enzymatic activity to an increased inflammatory phenotype driven by a TNF-α increase in later stages of MASLD [35]. Together, these findings show the interconnectedness between ethanol, liver inflammation and a decreased liver function to clear a high microbial ethanol formation from dietary sugars, adding to the pathogenesis and progression of MASLD-MASH.

### 2.2. Alterations in the Gut Microbiome in MASLD-MASH

Recent advances in sequencing technologies have made the study of compositional and functional studies a norm in microbiological studies [22,23,36]. These approaches have enabled various population studies to probe compositional and functional signatures of the GM in MASLD compared to healthy groups (Figure 1) [22,24,36,37], with some authors linking certain shifts to the liver fibrosis score [22,38]. At a broad taxonomic level, studies have reported an increase in the *Proteobacteria* phylum in fecal and luminal samples of MASLD patients, which is mainly driven by an increase in the *Enterobacteriaceae* family [22,24,37,39,40]. Other taxonomic phyla require complex analysis owing to the complexity and inherent diversity within those taxa. In many taxonomical studies, *Firmicutes* and *Bacteroidetes* are often regarded as a unit, despite the diverse metabolic capabilities and varying associations with MASLD among the genera within these groups [36,39]. A widely used index jointly assessing both phyla is the *Firmicutes* to *Bacteroidetes* ratio (F/B ratio). Although considerable variability has been reported across compositional studies, the prevailing trend indicates a relative enrichment of *Firmicutes* and a depletion of *Bacteroidetes* in individuals with MASLD compared to healthy controls [24,37]. Although the reasons for these inconsistencies are not fully elucidated, they may arise from the wide-ranging metabolic capacities of the constituent taxa. Importantly, the recognition of three distinct enterotypes in the human gut microbiome—each dominated by *Bacteroides*, *Prevotella*, or *Ruminococcus*—suggests that broad measures such as the F/B ratio may oversimplify community structure and provide limited insight into MASLD-associated alterations [24,39]. While reductions in the *Bacteroidetes* phylum are commonly reported in MASLD [24,39], in agreement with the F/B ratio, evidence suggests that an increase in the abundance of the species *Prevotella copri* has direct implications to MASLD progression [41] and inflammation in the gut via lipopolysaccharide (LPS) biosynthesis [42]. Although an overall increase in the *Firmicutes* phylum is reported in most MASLD-MASH compositional studies [24,37], notable genus-level discrepancies have been observed. Specifically, genera such as *Veillonellaceae* [22] and *Sarcina* [38] are described to increase in abundance, whereas beneficial genera including *Faecalibacterium* and *Ruminococcus* are consistently reduced [43,44]. Hence, although broad taxonomic analyses remain valuable for identifying shifts in microbial composition and their association with health implications, transitioning towards analysis with multi-omics approaches is essential to elucidate the specific metabolic and ecological processes that contribute to MASD-MASH pathogenesis.

To gain deeper insight into the functional roles of the gut microbiome, several studies have moved from compositional profiling towards metagenomic and metatranscriptomic approaches [25]. These analyses have revealed enhanced microbial degradation pathways for carbohydrate [24,25], lipid [25,40], and amino acid metabolism [25,40], accompanied by reductions in fatty acid, protein and SCFA biosynthesis in MASLD patients [42]. Nonetheless, not all studies are consistent in the reported changes and some authors attributing enhanced carbohydrate metabolization linked exclusively to later stages of MASLD [42]. Despite this variability, an enhanced carbohydrate metabolization is crucial to drive an increased endogenous ethanol synthesis that occurs in MASLD-MASH, although mechanistic details of this association require further clarification.

## 3. Pathways for Endogenous Ethanol Production

Ethanol biosynthesis has been extensively characterized in yeasts and certain bacteria, largely due to its relevance in industrial applications [45,46,47]. However, there is growing interest on endogenous ethanol production by the gut microbiota, particularly in relation to its potential role in MASLD [48]. Ethanol biosynthesis in the small and large intestine occurs during the fermentation of dietary sugars under anaerobic conditions with organic sources as the sole electron donors and acceptors [49]. Due to the importance of yeasts in auto-brewery syndrome, Mbaye et al. studied their implications in MASLD, finding fructose-dependent ethanol producing yeasts in 90% of their MASLD cohort and significantly more ethanol production than the bacterial counterparts [50,51]. However, other studies have not reported data on yeast abundance or reported no association with ethanol production [20]. Due to the higher abundance and significance of bacteria in the gut microbiome, the focus of endogenous ethanol production is shifted towards the bacterial population, where some isolates have already been identified as ethanol producers (Table 1).

Fermentative pathways in bacteria are common and diverse, with the synthesis of SCFAs and by-products, including ethanol (Figure 2). Fermentation routes are generally named by the main end-product, with 2,3-butanediol, mixed-acid, acetone–butanol and heterolactic fermentation being the known pathways to derive to ethanol synthesis in the gut environment [52,53]. Interestingly, compositional analysis have revealed that several ethanol-producing bacteria capable of fermenting dietary sugars are enriched in MASLD [19,20,26,54], highlighting the potential contribution of endogenous ethanol production to disease pathogenesis.

**Table 1 ijms-26-08609-t001:** Human bacterial isolates with ethanol-producing abilities.

Bacterial Strain	Original Isolation Source	Substrates	Pathway	References
*Bifidobacterium longum* BB536	Infant fecal sample, healthy	Fructose	Heterolactic	[55,56]
*Enterocloster bolteae*	Fecal sample, MASLD	Glucose, arabinose, fructose, sucrose, glycerol, maltose, mannose, melezitose, sorbitol, trehalose and xylose	Mixed acid	[57]
*Enterocloster bolteae* s28/42	Fecal sample, hepB	Glucose	Mixed acid	[58]
*Klebsiella pneumoniae* TH1	Fecal sample, MASH	Glucose and fructose	2,3-butanediol	[59]
*Klebsiella pneumoniae* W14	Fecal sample, MASH	Glucose and fructose	2,3-butanediol	[54,59]
*Lactobacillus fermentum* 92294	Food, residing in gut	Glucose	Heterolactic	[48]
*Lactobacillus reuteri* ATCC 6475	Breast milk	Glucose	Heterolactic	[60]
*Limosilactobacillus fermentum*	Fecal sample, MASH	Glucose	Heterolactic	[57]
*Streptococcus mutans*	Fecal sample, MASH	Glucose	n.s.	[57]
*Weissella confusa*	Fecal sample, healthy	Glucose, fructose	Heterolactic	[48,61]

hepB, hepatitis B patient; n.s., not specified.

For all metabolic pathways, fermentation starts with the breakdown of monosaccharides via glycolysis, which can occur via the classical Embden–Meyerhof–Parnas (EMP) pathway, the Entner–Doudoroff (ED) or the pentose–phosphate (PP) pathway [62]. The EMP and ED pathways yield energy in the form of ATP and NADH through the breakdown of glucose into pyruvate (Figure 2). However, the PP-pathway converts glucose into ribose sugars, yielding NADPH and metabolites used as cellular building blocks or energy formation [63]. The convergence of the three pathways into pyruvate, one of the central compounds for carbohydrate metabolism, allows for the synthesis of a great diversity of compounds, including SCFAs and alcohols. Bacterial routes able to derive in ethanol formation are the mixed-acid, heterolactic, 2,3-butanediol and acetone–butanol–ethanol (ABE) pathways. Nonetheless, it has been observed that growth and the metabolic byproducts of fermentative pathways are impacted by an acidic pH [64]. Changes in membrane permeability and protein synthesis brought on by the fermentation’s acidic byproducts, such acetate and lactate, have been observed to cause these alterations [64,65].

Both mixed-acid and heterolactic fermentative pathways are highly abundant in gut microbes and both lead to ethanol formation [49]. While the mixed-acid fermentation will derive in the production of a wider variety of SCFAs such as acetate, lactate and succinate, heterolactic fermentation generally only leads to lactate synthesis, with acetate as an additional bi-product in some cases (Figure 2) [49,66,67]. Mixed acid fermentation is widely present in *Enterobacteriaceae* genera, which includes some opportunistic and pathogenic microorganisms such as *Escherichia coli* and *Salmonella enterica* [68]. Meanwhile, heterolactic fermentation is mostly present in lactic acid bacteria, which are generally seen as beneficial microbes [67].

In the case of the 2,3-butanediol pathway, pyruvate is the common nexus between the formation of 2,3-butanediol and ethanol, with the balance being shifted towards 2,3-butanediol formation. This pathway is present in gut isolates such as *Klebsiella pneumoniae* and *Enterobacter* spp., which have been seen to be associated with MASLD [54,59,69,70]. Although the relevance of this pathway to ethanol formation is generally limited due to the minimal ethanol synthesis observed, its presence could be an important mediator in pathogenicity, as observed in the case of *Pseudomonas aeruginosa* in the lung environment [68]. These findings indicate a potential mechanism by which ethanol-producing gut bacteria could be associated to MASLD additional to that of ethanol formation.

The ABE fermentative pathway is divided in the acidogenic and solventogenic phase, with the first phase leading to the formation of acetate and butyrate, while the solventogenic phase enables the formation of ethanol, acetone and butanol [71,72,73]. Although most studies reporting this pathway have been focusing on the production of biofuels [74], some gut isolates of *Clostridiaceae* family are able to perform this pathway [67,73], raising the possible importance of this fermentative route in the gut while its prevalence has not been described.

Ethanol is formed via the conversion of acetyl-CoA into acetaldehyde by an aldehyde dehydrogenase (ALDH), which is eventually metabolized to ethanol by alcohol dehydrogenase (ADH) [52]. These enzymes conduct a dual role, as they have been related to both metabolism and pathogenicity in bacteria [75,76,77,78]. This is the case for enterohemorrhagic *E. coli*, organism which presents a type three secretion system that has been seen to be regulated by an ADH enzyme [79]. More recently, Lin et al. found impaired quorum sensing in the pathogenic *Acinetobacter baumanii* after ADH inhibition, leading to decreased biofilm formation and motility relevant for its pathogenic implications [80]. In addition, in the pathogenic *Bacillus cereus* an ADH subtype (AdhB) functions as a virulence factor and is implicated in oxidative and nitric stress [81]. Collectively, these studies highlight the multifunctionality of ADH enzymes in pathogenic bacteria, suggesting additional mechanisms by through ethanol-producing microorganisms can influence small and large intestinal microbial homeostasis and host health.

## 4. Ethanol as a Substrate for Microbial Metabolism

While most research has focused on the microbially produced ethanol, evidence from the field of environmental microbiology indicates that certain anaerobic bacteria have the capacity to utilize ethanol as a substrate, particularly in the presence of external terminal electron acceptors [82]. The presence of ethanol-oxidizing microorganisms in the gut environment in MASLD patients could add to the clearance of ethanol from the liver and positively contribute to the disease progression [32,83]. However, certain potentially pathogenic species have been shown to be able to accumulate the toxic intermediate acetaldehyde in the human colon [83] and oral cavity [84], thereby exacerbating the negative effects associated with ethanol synthesis and metabolism. Therefore, the efficient microbial oxidation of ethanol into acetate or CO_2_ within a healthy microbiome is critical for mitigating the detrimental effects of ethanol. In the gastrointestinal environment and beyond.

There is limited data regarding the mechanisms by which gut microbial strains can utilize ethanol, and some researchers have questioned whether ethanol metabolization can occur under gut conditions both due to intestinal pH as well as unfavorable thermodynamics of the reaction in the absence of oxygen [85,86]. Although the intestinal tract is predominantly anaerobic, the presence of oxygen in the small intestine, distal colon, along with the epithelial-luminal gradient, and the availability of alternative electron acceptors such as nitrate and sulfate, suggests that ethanol oxidation could occur in the gut through respiration coupled with various electron acceptors [87,88,89].

Utilization of ethanol in bacteria involves its oxidation to acetaldehyde via an ADH enzyme [90,91]. This compound functions as the intermediate of the reaction and is converted to acetate in a direct or indirect manner (Figure 3) [91,92]. The direct conversion involves a non-acetylating acetaldehyde dehydrogenase (ALDH) or an aldehyde ferredoxin oxidoreductase (AOR) that result in the oxidation of acetaldehyde to acetate [90,91]. Contrarily, the indirect conversion of acetaldehyde will involve an acetylating ALDH enzyme that is able to incorporate a cofactor A (CoA) into acetaldehyde and synthesize acetyl-CoA [91,93]. This central metabolite can either be used in multiple cellular metabolic pathways or converted to acetate. The conversion into acetate occurs via an acetyl-CoA synthetase (ACS) in a single reaction, or in a two-step pathway involving a phosphate acetyltransferase (PTA) and an acetate kinase (ACK). Both pathways allow for the formation of ATP in the formation of acetate [91,94], yielding energy to the cell additional to that obtained in the recovery of the NAD^+^/NADH pool. Due to the nature of ethanol, a 2 carbon molecule, its metabolization is associated with a lower biomass production and slowed growth rate compared to simple sugars [91].

While the reaction can take place in anoxic conditions, it has been described that the presence of oxygen allows for a more efficient ethanol oxidation, rooted in the electron and redox balance [83,84]. However, alternative electron acceptors can be used in the anoxic oxidation of ethanol, such as the sulfate, nitrate and others [83,88,95]. In environmental microbiology, it is known that sulfate-reducing bacteria (SRB) of the genus Desulfovibrio are able to oxidize ethanol and couple it with sulfate reduction, resulting in the production of hydrogen sulfide or reduced sulfide, sulfate or sulfur ions [87,95,96]. Nevertheless, it remains unclear whether intestinally residing Desulfovibrio species are capable of performing similar ethanol oxidation coupled to sulfate reduction. In the case of nitrate-reducing bacteria (NRB), there evidence is limited on their ability to oxidize ethanol by using nitrate as the terminal electron acceptor in the gut [97]. It is important to consider that nitrate follows a gradient along the gastrointestinal tract, with higher concentrations present in the small intestinal lumen and lower levels in the colon [98]. This spatial gradient of nitrate availability may affect the distribution and metabolic activity of nitrate-reducing bacteria through the gut [98]. However, limited data supports the occurrence of ethanol oxidation coupled with nitrate reduction in the gut, suggesting that this metabolic pathway may may be less prevalent or significant than sulfate reduction. Further research is needed to clarify the role of nitrate-reducing bacteria and the use of additional electron acceptors depicted in Figure 3 in ethanol metabolism within the intestinal environment and to understand their potential implications for gut health.

Numerous taxonomical groups have been reported to possess bacterial ADH and ALDH enzymes capable of oxidizing ethanol under both aerobic and anaerobic conditions. For example, bacteria from the *Neisseria* spp. and *Streptococcus* spp. in the oral microbiome have been shown to express ADH enzymes that catalyze the oxidation of ethanol to acetaldehyde and acetate [84]. Additionally, Tsuruya et al. isolated several aerobic, anaerobic and facultative anaerobic bacteria from the human gut capable of oxidizing ethanol to acetaldehyde [83]. Additional human and environmental ethanol oxidizing bacteria can be found in Table 2.

## 5. Gut Bacteria as a Therapeutic Target for MASLD-MASH

Although the precise pathophysiological mechanisms underlying of MASLD remain incompletely defined, accumulating evidence indicates the gut microbiome as a critical factor in disease onset and progression. Current treatment options for MASLD patients focuses on lifestyle interventions and addressing underlaying comorbidities [27,103]. The absence of specific targeted treatments has driven research into microbiome-focus interventions as potential therapies due to the strong implications of the gut microbiome in MASLD pathogenesis. Microbiome-based approaches can be focused on the use of microorganisms in the form of probiotics and prebiotics, or on heterologous fecal microbial transplantations. Additionally, considering the importance of ethanol production on MASLD, other approaches could focus on targeting the microbial ethanol production (Figure 4). However, up to date no research has been performed on this approach.

Fecal microbial transplantations (FMTs) from healthy donors, referred to as heterologous FMT, show great potential as microbiome-based intervention in MASLD, with effective reduction of body weight, fasting blood glucose, triglyceride levels and liver markers [104,105]. However, no human interventions have been completed on the effectiveness of FMT in MASLD up to date. Another avenue taken is the use of an individually developed microbial consortia or synthetically developed FMTs which supplement the depletion of beneficial microbes in MASLD patients. This is the case of Kwan et al., who used a consortium of microorganisms which were seen to be depleted in their MASH cohort in a diet-induced MASLD mouse model [106]. Their results show a protection for the MASLD liver pathogenesis in the mice, which points towards the beneficial role of personalized supplementation in MASLD patients.

The use of the traditional and most well-studied probiotic bacteria is another route some authors explore. These microorganisms are generally included in the Bifidobacterium and *Lactobacillus* genus and have proven to have beneficial effects in the gut microenvironment in a broad number of cases [107]. An example of this is the fecal isolate *Bifidobacterium longum* subsp. *longum* BL21 (BL21), which was able to attenuate ethanol-induced intestinal inflammation and ameliorate ethanol-induced liver steatosis and oxidative stress in mice [108]. The same strain was studied in the context of T2D in mice, leading to an ameliorated glucose metabolism and reduced hyperglycemic-induced liver damage, while also modulating the GM towards a decreased F/B ratio [109]. Another example of potential probiotic is the lactic acid bacteria *Lactobacillus helveticus*. This microorganism has been found to mitigate ethanol-induced liver damage via alterations in the gut microbial composition, SCFA production and intestinal permeability [110]. Although these examples show potential in in vivo models, clinical trials are required to draw conclusions on the relevance of BL12 and *L. helveticus* supplementation in MASLD patients. Combining well-researched bacterial probiotics or a customized bacterial consortia with prebiotics could provide a synergistic effect [107]. This approach was adopted in a randomized controlled trial by Mitrović et al., based on the combination of *Bifidobacterium lactis*, *Lactobacillus* spp. and inulin [111]. Their results highlight the potential of pro- and prebiotics on liver fat accumulation and in shaping the GM composition. This strategy allows for a personalized approach and microbiome-based therapies to target specific alterations, posing great potential for ethanol-targeting approaches in MASLD patients.

As endogenous ethanol production has been proven to be one of the drivers of MASLD [20,26,57], one of the potential approaches to be further investigated includes the chemical inhibition of ethanol synthesis in the gastrointestinal tract. Although no data has reported on the inhibition of bacterial ADH enzymes in vivo, Zetterström et al. investigated the in vitro inhibition of the AdhE enzyme in enterohemorrhagic *E. coli* as a strategy to reduce strain pathogenicity. Their work demonstrated successful chemical inhibition of the ADH-mediated conversion of acetyl-CoA to ethanol, although no evidence was provided for a corresponding reduction in pathogenicity [79]. Importantly, authors that seek to study this approach ought to consider potential off-target effects both the host and the gut microbiome. In the host, compromising the activity of the ADH enzymes represents major safety concerns due to the implications for ethanol clearance and associations to MASLD-MASH [32]. Potential impacts on microbial metabolic profiles, the accumulation of toxic intermediates of ethanol metabolism [112], and disruptions to ecological balance require careful consideration. These potential side-effects that come from chemical enzyme inhibition could be major setbacks in the progress of research performed in targeting the enzymes responsible for ethanol formation in gut bacteria. However, these authors believe that the specific chemical inhibition of bacterial ADH could be a promising approach to mitigating the formation of endogenous ethanol in the gut and managing MASLD-MASH more effectively.

## 6. Conclusions and Future Perspectives

The current knowledge on the implication of the gut microbiome in the pathogenesis of MASLD and the link with disease onset and progression are unclear. However, it is known that MASLD is accompanied by a shift in the gut microbial composition and recent studies have described and provided enough evidence to point to the importance of endogenous ethanol biosynthesis [23,25,57]. However, no clinical studies have been performed to study the causal relationship between the two, and more studies need to be performed to fully understand how the microbiome is altered in MASLD and how it derives to a high ethanol biosynthesis in the gut.

Evidence points towards the potential therapeutic effects of interventions targeting the gut microbiome, such as administration of probiotic, prebiotics, or FMTs. These are showing potential in the studies previously described but require further studying to be able to bring them to the clinic. Alternative and new strategies targeting the altered ethanol metabolism in the gut could also prove to be beneficial for balancing the gut microbiome composition and reducing the negative effects ethanol biosynthesis has in the gut and liver. Hence, the authors suggest that microbiome-targeting therapies be developed and added to the currently-in-place lifestyle interventions for the management of MASLD, a disease with increasing prevalence in society.

## Figures and Tables

**Figure 1 ijms-26-08609-f001:**
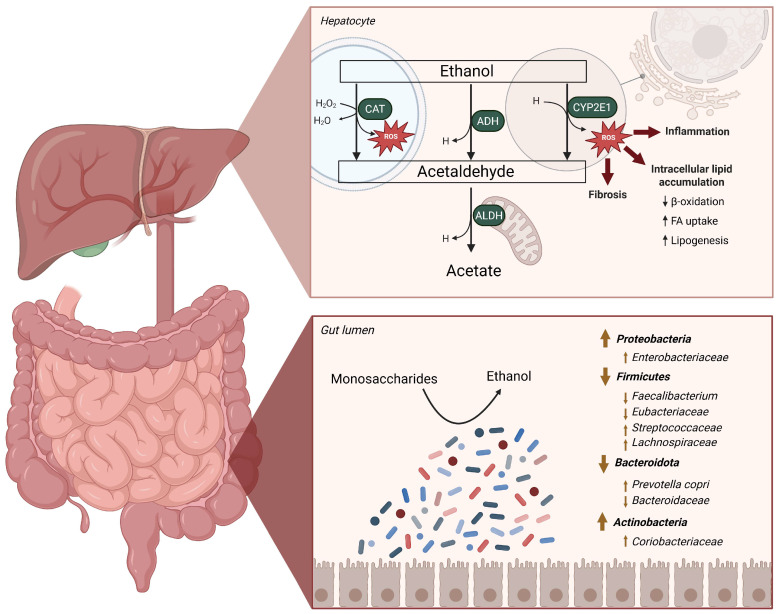
Metabolism of endogenous ethanol in the liver and alterations in the gut microbiome in MASLD-MASH. The gut microbiome is able to ferment dietary sugars such as glucose or fructose into ethanol, which is transported via the portal vein into the liver. When it reaches the hepatocytes, ethanol undergoes degradation via three pathways: through the catalase system (CAT) located in the peroxisome, in the cytosol via the alcohol dehydrogenase (ADH) enzyme, or in the endoplasmic reticulum (ER) via the cytochrome P450 2E1 (CYP2E1) system. The product of all three reactions, acetaldehyde, is converted via an acetaldehyde dehydrogenase (ALDH) located in the mitochondria, into acetate. The formation of reactive oxygen species (ROS) in the hepatocytes due to the degradation of ethanol via the CAP and CYP2E1 routes added to the presence of acetaldehyde, leads to fibrosis, inflammation, and intracellular lipid accumulation via alterations of the fatty acid (FA) metabolism, decreased β-oxidation and to increased lipogenesis. In the bottom part of the figure the main microbial alterations associated with MASLD are seen. ADH, alcohol dehydrogenase; ALDH, acetaldehyde dehydrogenase; CAT, catalase; CYP2E1, cytochrome P450 2E1; FA, fatty acids; H, reduced nicotinamide adenine dinucleotide; ROS, reactive oxygen species.

**Figure 2 ijms-26-08609-f002:**
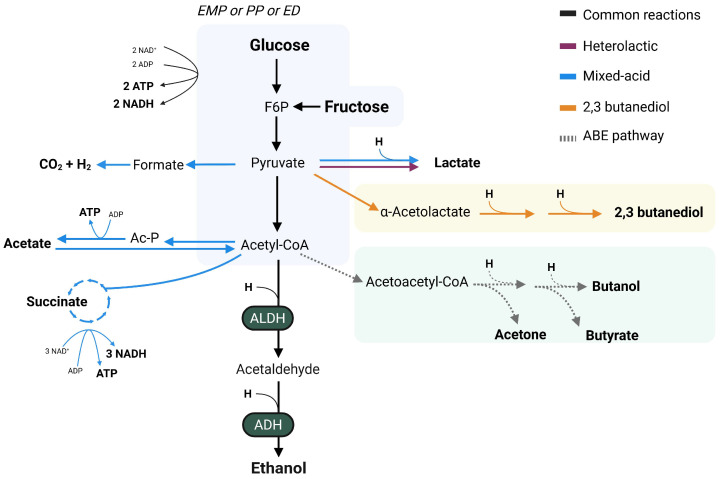
Overview of fermentative pathways involved in ethanol biosynthesis. The synthesis of ethanol by gut microbes can occur via different fermentative pathways which start with the glycolysis of simple sugars via the Embden–Meyerhof–Parnas pathway (EMP), the Entner–Doudoroff pathway (ED) or the pentose–phosphate pathway (PP). The synthesis of ethanol involves the conversion of glycolysis-produced acetyl-CoA into ethanol via the intermediate acetaldehyde through the enzymes acetaldehyde dehydrogenase (ALDH) and alcohol dehydrogenase (ADH). Heterolactic fermentation (purple) results in the formation of lactate and ethanol, while mixed-acid fermentation (blue) results in the production of lactate, acetate and the intermediates formate and succinate. The 2,3-butanediol-pathway (yellow) yields the synthesis of 2,3-butanediol and ethanol from pyruvate. The acetone–butanol–ethanol (ABE) (grey, dashed) pathway leads to the formation of acetone, butanol, butyrate and ethanol. Common reactions in all metabolic pathways are shown in black, while the specific reaction part of the pathways are shown in their respective colors—purple for heterolactic, blue for mixed-acid, orange for the 2,3-butanediol and grey for ABE pathway. Solid lines indicate the most common pathways in bacteria, while dashed lines show not very commonly described pathways. AcP, acetyl phosphate; ADH, alcohol dehydrogenase; ADP, adenosine diphosphate; ALDH, aldehyde dehydrogenase; ATP, adenosine triphosphate; ED, Entner-Doudoroff pathway; EMP, Embden–Meyerhof–Parnas pathway; F6P, fructose-6-phosphate; H, reduced nicotinamide adenine dinucleotide; PP, pentose–phosphate pathway.

**Figure 3 ijms-26-08609-f003:**
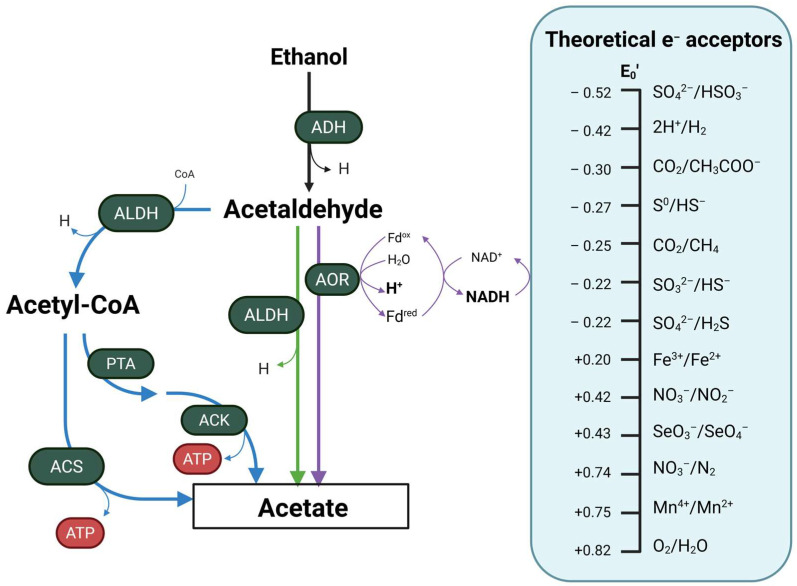
Bacterial ethanol oxidation pathways. Ethanol is oxidated via and alcohol dehydrogenase (ADH) into acetaldehyde, which is further directly converted to acetate via an aldehyde ferredoxin oxidoreductase (AOR) or non-acetylating acetaldehyde dehydrogenase (ALDH). The conversion of acetaldehyde to acetate can also occur via the formation of acetyl-CoA, involving a non-acetylating acetaldehyde dehydrogenase, a phosphate acetyltransferase (PTA) and acetate kinase (ACK) or an acetyl-CoA synthetase (ACS). These reactions can be coupled with a diversity of external electron acceptors under anaerobic conditions that are used to recycle the NAD^+^/NADH balance inside the cell. These redox couples are arranged in order from most electronegative E_0_′ (top) to most electropositive (bottom) assuming neutral pH and anaerobic conditions. ACK, acetate kinase; ACS, acetyl-CoA synthetase; ADH, alcohol dehydrogenase; ALDH, acetaldehyde dehydrogenase; AOR aldehyde ferredoxin oxidoreductase; ATP, adenosine triphosphate; H, reduced nicotinamide adenine dinucleotide; NADH, reduced nicotinamide adenine dinucleotide; NAD, nicotinamide adenine dinucleotide; PTA, phosphate acetyltransferase.

**Figure 4 ijms-26-08609-f004:**
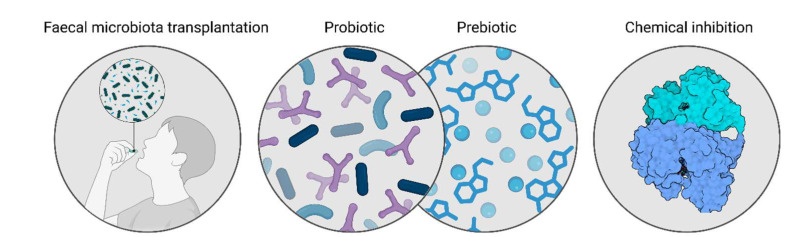
Potential microbiome-based approaches to target ethanol synthesis in MASLD. Microbiome-based approaches can target the microbial composition and focus on fecal microbiome transplantations (FMTs), use probiotic strains to shift the microbial balance and decrease the synthesis of ethanol and combine them with prebiotic substances that can enhance their effect. Moreover, additional strategies could target the alcohol dehydrogenase enzymes in the formation of ethanol via chemical inhibitors.

**Table 2 ijms-26-08609-t002:** Human and environmental isolates with ethanol-degrading abilities.

Bacterial Strain	Origin of Isolation	Reaction	e^−^ Acceptor	Oxygen Tolerance	References
*Acetobacterium woodii*	Mud	EtOH → Acetyl-CoA	CO_2_	Anaerobe	[99]
*Collinsella* spp.	Fecal	EtOH → AcH	O_2_	Aerotolerant	[83]
*Escherichia coli*	Fecal	EtOH → AcH	O_2_	Facultative anaerobe	[100]
*Methanogenium organophilum*	Sewage	EtOH → Ac	Sulfide	Anaerobe	[101]
*Neisseria mucosa*	Oral	EtOH → AcH	O_2_	Aerobe	[84]
*Neisseria sicca*	Oral	EtOH → AcH	O_2_	Microaerophile	[84]
*Pseudomonas aeruginosa* PA14	Wound	EtOH → Ac	O_2_ and nitrate	Facultative anaerobe	[86]
*Ruminococcus* spp.	Fecal	EtOH → AcH	n.s.	Obligate anaerobe	[83]
*Streptococcus mitis*	Oral	EtOH → AcH	O_2_	Microaerophile	[84]
*Streptococcus salivarius*	Oral	EtOH → AcH	O_2_	Microaerophile	[84]
*Syntrophotalea carbinolica*	Sludge	EtOH → Ac	Fe(III)	Facultative anaerobe	[102]

Ac, acetate; AcH, acetaldehyde; EtOH, ethanol; n.s., not specified.

## Data Availability

No new data were created or analyzed in this study. Data sharing is not applicable to this article.

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
