# Peer review of "Endogenous Ethanol Metabolism and Development of MASLD-MASH"

_ijms, 2025, doi:10.3390/ijms26178609_

Round 1
Reviewer 1 Report
Comments and Suggestions for Authors
I
Recommendation: accept in present form
Comments:
In this manuscript, the review article,which the title :“Endogenous ethanol metabolism and development of MASLD- 2 MASH” could be acceptable in present form, and no needed to be revised further.

Author Response
Comment 1: In this manuscript, the review article, which the title :“Endogenous ethanol metabolism and development of MASLD- 2 MASH” could be acceptable in present form, and no needed to be revised further.
Response 1: We sincerely thank the reviewer for their positive evaluation and supportive feedback. We are pleased that the manuscript is considered acceptable in its current form. Please find below the new version of the document in accordance to the other reviewer's comments. We sincerely hope this changes are in line with your expectations.

Reviewer 2 Report
Comments and Suggestions for Authors
Farràs Solé et al. discuss the role of endogenous ethanol production and metabolism in the development of metabolic-associated fatty liver disease (MASLD) and metabolic-associated steatohepatitis (MASH). The main focus is on gut microbiome alterations leading to impairments of ethanol synthesis and clearance, which may act as a causal factor in the pathogenesis of MASLD/MASH, and the potential microbiome-targeting treatments. The findings of the review are of considerable significance as they explore the most prominent novel therapeutic approaches for MASLD/MASH.
Comments:
- Correlations have been established between disease progression in MASLD/MASH and some bacterial taxa abundance besides the mentioned Prevotella; e.g., Bifidobacterium spp., Lactobacillus spp., Clostridium spp., Sarcina spp., Akkermansia muciniphila. At least some of the most prominent species/genera should be given a mention along with Prevotella spp. when discussing the major gut microbiome alterations in MASLD;
- Direct chemical inhibition of ethanol production is listed among the current approaches to MASLD therapy despite having absolutely no experimental evidence, as stated by the authors themselves. This approach requires either a relevant source for its theoretical basis (if it has been already proposed elsewhere), or a more detailed justification by the authors if it is their own original concept;
- Names of all bacterial taxa in both the text and figures should be italicized as per the current International Code of Nomenclature of Prokaryotes recommendations, while strain designations should not be italicized.
Author Response
We would like to thank the reviewer for their time and effort in the provided comments. Please find our answers and modifications in the message.
Comment 1: Correlations have been established between disease progression in MASLD/MASH and some bacterial taxa abundance besides the mentioned Prevotella; e.g., Bifidobacterium spp., Lactobacillus spp., Clostridium spp., Sarcina spp., Akkermansia muciniphila. At least some of the most prominent species/genera should be given a mention along with Prevotella spp. when discussing the major gut microbiome alterations in MASLD;
Response 1: We sincerely thank the reviewer for this constructive suggestion. We agree with the benefits in the addition of more specific alterations in gut microbes. We have added several that can be found in section 2.2. Please find it in page number 6, section 2.2 lines 156 to 159 and below.
“[…] While reductions in the Bacteroidetes phylum are commonly reported in MASLD [24, 40], in agreement with the F/B ratio, evidence suggests that an increase in the abundance of the species Prevotella copri has direct implications to MASLD progression [43] and inflammation in the gut via lipopolysaccharide (LPS) biosynthesis [44]. Although an overall increase in the Firmicutes phylum is reported in most MASLD-MASH compositional studies [24, 38], notable genus-level discrepancies have been observed. Specifically, genera such as Veillonellaceae [25] and Sarcina [39] are described to increase in abundance, whereas beneficial genera including Faecalibacterium and Ruminococcus are consistently reduced [25, 38]. Hence, although broad taxonomic analyses remain valuable for identifying shifts in microbial composition and their association with health implications, transitioning towards analysis with multi-omics approaches is essential to elucidate the specific metabolic and ecological processes that contribute to MASD-MASH pathogenesis.”
Comment 2: Direct chemical inhibition of ethanol production is listed among the current approaches to MASLD therapy despite having absolutely no experimental evidence, as stated by the authors themselves. This approach requires either a relevant source for its theoretical basis (if it has been already proposed elsewhere), or a more detailed justification by the authors if it is their own original concept;
Response 2: Thank you for the suggestion. As stated in the comment, the lack of evidence in the field makes it challenging to convey the message with enough evidence. Nevertheless, it is definitely worth investigating this approach in the future due to it high potential therapeutic applicability. It is for this reason that we have included in the justification a paper describing the screening for alcohol dehydrogenase inhibitors due to its importance in the mediation of pathogenicity of E. coli. We have included this information and expressed the need for careful consideration when studying this approach. The modifications can be found on page 13, section 5, lines 405 to 410.
“As endogenous ethanol production has been proven to be one of the drivers of MASLD [21, 27, 59], one of the potential approaches to be further investigated includes the chemical inhibition of ethanol synthesis in the gastrointestinal tract. Although no data has reported on the inhibition of bacterial ADH enzymes in vivo, Zetterström, et al. investigated the in vitro inhibition of the AdhE enzyme in enterohemorrhagic E. coli as a strategy to reduce strain pathogenicity. Their work demonstrated successful chemical inhibition of the ADH-mediated conversion of acetyl-CoA to ethanol, although no evidence was provided for a corresponding reduction in pathogenicity [82]. Importantly, authors that seek to study this approach ought to consider potential off-target effects both the host and the gut microbiome […]”
Comment 3: Names of all bacterial taxa in both the text and figures should be italicized as per the current International Code of Nomenclature of Prokaryotes recommendations, while strain designations should not be italicized.
Response 3: We would like to thank the reviewer for this highlight. We have made the required adjustments in accordance to the International Code of Nomenclature of Prokaryotes through the text and in Figure 1 (page 5, line 115). Changes can be found in section 2.2 (page 5-6) in lines 136, 137, 139, 140, 144, 148, 152-156, section 3 page 7 lines 224 and 225 and page 8 line 261 and section 5 page 12 lines 384
Please see the attachment for the revised version of the document.

Reviewer 3 Report
Comments and Suggestions for Authors
The manuscript offers a comprehensive and well-structured review of the role of endogenous ethanol production by the gut microbiome in the pathogenesis and progression of MASLD–MASH. Below is the comments:
- Section 2.2 The discussion on the Firmicutes/Bacteroidetes ratio is somewhat generalized. Including a nuanced perspective on its context-dependency and methodological variability across studies would add value.
- Figures 2 and 3, although informative, are visually dense. They could benefit from simplified layouts or highlighting key pathways relevant to MASLD pathogenesis rather than all possible fermentation/oxidation routes.
- Table 1 and Table 2: Good inclusion of organisms, but the “n.s.” (not specified) entries reduce utility; consider adding substrate/pathway info from related literature if possible.
- Abbreviations and terms such as “(small) intestine” and “MASLD-MASH” appear inconsistently — standardizing them would improve readability.
- Section 3: The emphasis on yeast-derived ethanol in MASLD is appreciated, but more discussion on why bacterial ethanol production is given primary focus despite yeast showing higher production in some cohorts would be useful.
- Section 5: The suggestion to inhibit ethanol synthesis enzymatically is novel but speculative — please elaborate on potential molecular targets and off-target effects would strengthen this proposal.
Author Response
We sicerely thank the reviewr for their time and insightful comments. We have updated the manuscipt accordingly, please see the modifications in our responses and attached document.
Comment 1: Section 2.2 The discussion on the Firmicutes/Bacteroidetes ratio is somewhat generalized. Including a nuanced perspective on its context-dependency and methodological variability across studies would add value.
Response 1: The point made by reviewer 3 has been carefully considered and we agree that a more clarified explanation on the F/B ratio would be beneficial for the reader. We have added an explanation that we believe is satisfactory in understanding the message to convey. Please find the revised version in Section 2.2, page 5, lines 145 to 151.
“[…] A widely used index jointly assessing both phyla is the Firmicutes to Bacteroidetes ratio (F/B ratio). Although considerable variability has been reported across compositional studies, the prevailing trend indicates a relative enrichment of Firmicutes and a depletion of Bacteroidetes in individuals with MASLD compared to healthy controls [24, 38]. Although the reasons for these inconsistencies are not fully elucidated, they may arise from the wide-ranging metabolic capacities of the constituent taxa. Importantly, the recognition of three distinct enterotypes in the human gut microbiome—each dominated by Bacteroides, Prevotella, or Ruminococcus—suggests that broad measures such as the F/B ratio may oversimplify community structure and provide limited insight into MASLD-associated alterations [42]. […]”
Comment 2: Figures 2 and 3, although informative, are visually dense. They could benefit from simplified layouts or highlighting key pathways relevant to MASLD pathogenesis rather than all possible fermentation/oxidation routes.
Response 2: Thank you for the comment, we agree Figure 2 could indeed benefit from simplification in some steps. We have simplified the NAD/NADH conversion steps and glycolysis. Additionally, the key fermentative pathways have been highlighted. Please find the revised version on page 8, line 238. In figure 3, we have also modified the layout to make it more attractive to the reader and maintained the same style of simplification as figure 2 (page 10, line 306). That is, the simplification of NAD/NADH conversions. Additionally, as to maintain cohesiveness among the text, the simplification of the NAD/NADH conversion has been added to figure 1. Please find the revised figures in pages 4, 8 and 10.
Comment 3: Table 1 and Table 2: Good inclusion of organisms, but the “n.s.” (not specified) entries reduce utility; consider adding substrate/pathway info from related literature if possible.
Response 3: These authors agree that additional information in the tables is beneficial. However, we risk overgeneralization when it comes to the metabolic abilities of specific strains. Hence, we have added several pathways in Table 1 of those strains or species were solid evidence exists on their metabolic ability from experimental data. Please find the revised version on page 6, line 197. As for table 2, likely electron acceptors have been added to the table based on known electron acceptor usage under other experimental conditions not related to ethanol oxidation, thus completing the missing data. Please find the revised version on page 11, line 350.
Comment 4: Abbreviations and terms such as “(small) intestine” and “MASLD-MASH” appear inconsistently — standardizing them would improve readability.
Response 4: We appreciate the highlight, we have adjusted the term (small) intestine for “small and large intestine”. Please see page 3, line 90, page 5 line 177 and page 9 line 269. As for MASLD-MASH, we have updated some inconsistencies in page 4 line 130, and page 5, line 170.
Comment 5: Section 3: The emphasis on yeast-derived ethanol in MASLD is appreciated, but more discussion on why bacterial ethanol production is given primary focus despite yeast showing higher production in some cohorts would be useful.
Response 5: Thank you for the remark. We have adjusted the text so that it is clear to the reader why we shift our focus to bacterially-derived ethanol; the reasons being little evidence in microbiome studies in the field of MASLD-MASH on the relevance of yeasts to the overall composition, and most studies on ethanol formation by yeasts being from industrial research. Please find the modifications in section 3, page 6 lines 182-184.
“ […]Ethanol biosynthesis in the small and large intestine occurs during the fermentation of dietary sugars under anaerobic conditions with organic sources as the sole electron donors and acceptors [51]. Due to the importance of yeasts in auto-brewery syndrome, Mbaye, et al. studied their implications in MASLD, finding fructose-dependent ethanol producing yeasts in 90% of their MASLD cohort and significantly more ethanol production than the bacterial counterparts [51, 52]. However, other studies have not reported data on yeast abundance or indicated no association with ethanol production [21]. Hence, due to the higher abundance and significance of bacteria in the gut microbiome, the focus of endogenous ethanol production is shifted towards the bacterial population, where some isolates have already been identified as ethanol producers (Table 1).”
Comment 6: Section 5: The suggestion to inhibit ethanol synthesis enzymatically is novel but speculative — please elaborate on potential molecular targets and off-target effects would strengthen this proposal.
Response 6: We thank the reviewer for the useful suggestion. We have added additional justification to the section discussing chemical inhibition of alcohol dehydrogenase enzymes regarding this aspect. The modifications can be found on pages 13, section 5, lines 410-421.
“As endogenous ethanol production has been proven to be one of the drivers of MASLD [21, 27, 59], one of the potential approaches to be further investigated includes the chemical inhibition of ethanol synthesis in the gastrointestinal tract. Although no data has reported on the inhibition of bacterial ADH enzymes in vivo, Zetterström, et al. investigated the in vitro inhibition of the AdhE enzyme in enterohemorrhagic E. coli as a strategy to reduce strain pathogenicity. Their work demonstrated successful chemical inhibition of the ADH-mediated conversion of acetyl-CoA to ethanol, although no evidence was provided for a corresponding reduction in pathogenicity [82]. Importantly, authors that seek to study this approach ought to consider potential off-target effects both the host and the gut microbiome. In the host, compromising the activity of the ADH enzymes represents major safety concerns due to the implications for ethanol clearance and associations to MASLD-MASH [33]. Potential impacts on microbial metabolic profiles, the accumulation of toxic intermediates of ethanol metabolism [115], and disruptions to ecological balance require careful consideration. These potential side-effects that come from chemical enzyme inhibition could be major setbacks in the progress of research performed in targeting the enzymes responsible for ethanol formation in gut bacteria. However, these authors believe that the specific chemical inhibition of bacterial ADH could be a promising approach to mitigating the formation of endogenous ethanol in the gut and managing MASLD-MASH more effectively.”
